# Effect of Multiply Twinned Ag^(0)^ Nanoparticles on Photocatalytic Properties of TiO_2_ Nanosheets and TiO_2_ Nanostructured Thin Films

**DOI:** 10.3390/nano12050750

**Published:** 2022-02-23

**Authors:** Snejana Bakardjieva, Jakub Mares, Eva Koci, Jakub Tolasz, Radek Fajgar, Vasyl Ryukhtin, Mariana Klementova, Štefan Michna, Hana Bibova, Randi Holmestad, Rositsa Titorenkova, Maria Caplovicova

**Affiliations:** 1Institute of Inorganic Chemistry of the Czech Academy of Sciences, 250 68 Husinec-Rez, Czech Republic; mares@iic.cas.cz (J.M.); evulekotule@iic.cas.cz (E.K.); jtolasz@iic.cas.cz (J.T.); 2Faculty of Mechanical Engineering, JE Purkyně University, Pasteurova 1, 400 96 Ústí nad Labem, Czech Republic; stefan.michna@ujep.cz; 3Institute of Chemical Process and Fundamentals of the Czech Academy of Sciences, Rozvojova 2/135, 165 02 Prague, Czech Republic; fajgar@icpf.cas.cz; 4Nuclear Physics Institute of Czech Academy of Sciences, 250 68 Husinec-Rez, Czech Republic; ryukhtin@ujf.cas.cz; 5Institute of Physics of the Czech Academy of Sciences, Na Slovance 1999/2, 182 21 Prague, Czech Republic; klemari@fzu.cz; 6J. Heyrovsky Institute of Physical Chemistry of the Czech Academy of Sciences, Dolejskova 2155/3, 182 23 Prague, Czech Republic; hana.bibova@jh-inst.cas.cz; 7Department of Physics, Norwegian University of Science and Technology (NTNU), NO 7491 Trondheim, Norway; randi.holmestad@ntnu.no; 8Institute of mineralogy and crystallography, Bulgarian Academy of Sciences, 107, Acad. G. Bonchev Str., 1113 Sofia, Bulgaria; rositsatitorenkova@imc.bas.bg; 9STU Centre for Nanodiagnostics, University Science Park Bratislava Centre, Slovak University of Technology, Vazovova 5, 811 07 Bratislava, Slovakia; maria.caplovicova@stuba.sk

**Keywords:** anatase, Ag^(0)^ NPs, twinned defects, nanosheets, thin films, photocatalytic activity

## Abstract

Ag-decorated TiO_2_ nanostructured materials are promising photocatalysts. We used non-standard cryo-lyophilization and ArF laser ablation methods to produce TiO_2_ nanosheets and TiO_2_ nanostructured thin films decorated with Ag nanoparticles. Both methods have a common advantage in that they provide a single multiply twinned Ag^(0)^ characterized by {111} twin boundaries. Advanced microscopy techniques and electron diffraction patterns revealed the formation of multiply twinned Ag^(0)^ structures at elevated temperatures (500 °C and 800 °C). The photocatalytic activity was demonstrated by the efficient degradation of 4-chlorophenol and Total Organic Carbon removal using Ag-TiO_2_ nanosheets, because the multiply twinned Ag^(0)^ served as an immobilized photocatalytically active center. Ag-TiO_2_ nanostructured thin films decorated with multiply twinned Ag^(0)^ achieved improved photoelectrochemical water splitting due to the additional induction of a plasmonic effect. The photocatalytic properties of TiO_2_ nanosheets and TiO_2_ nanostructured thin films were correlated with the presence of defect-twinned structures formed from Ag^(0)^ nanoparticles with a narrow size distribution, tuned to between 10 and 20 nm. This work opens up new possibilities for understanding the defects generated in Ag-TiO_2_ nanostructured materials and paves the way for connecting their morphology with their photocatalytic activity.

## 1. Introduction

In 2016, the World Health Organization (WHO) reported that air pollution occupied the sixth position among all factors leading to death globally [1]. Heterogeneous photocatalysis based on titania (TiO_2_) and modified titania is believed to be an appropriate approach with a great capacity for improving both air and water quality, and generally refining human health. TiO_2_ is accepted as one of the most effective photo-induced catalysts used to destroy pollutants in our environment due to its strong redox ability and photostability. TiO_2_ is known to be a non-expensive, nontoxic material that exists in three polymorphic structures—anatase, rutile, and brookite. Among these three polymorphs, anatase has attracted the greatest interest in photocatalysis. It is commonly accepted that its photocatalytic activity is affected by light absorption, charge creation, charge mobility, and charge recombination rate, as well as surface reactivity. On the other hand, the direct wide bandgap (*E_bg_*) of anatase (3.2 eV) remains a major obstacle, limiting its solar energy utilization to about 4% [2]. In this context, significant efforts have been directed toward improving the catalytic activity of TiO_2_ by modifying its *E_bg_*. Generally, nanostructured photocatalysts [3] have been demonstrated to show significant promise, and provide better performance in catalyzing reactions due to their distinct physicochemical properties (melting point, wettability, electrical and thermal conductivity, light absorption, and scattering). These advantages are mainly a result of their nanoscale size (according to the EU Commission, this is defined as particles with external dimensions in the size range between 1 and 100 nm) [4] and structure-dependent properties that are distinct from their bulk counterparts [5]. In the last two decades, hundreds of novel nanostructured TiO_2_ materials have been developed. The classification scheme includes zero-dimensional (0D), one-dimensional (1D), two-dimensional (2D), and three-dimensional (3D) TiO_2_ nanomaterials. The number and variety of TiO_2_ belonging to the 2D family are far lower than for 0D and 1D morphologies. 2D TiO_2_ assemblies such as nanosheets usually do not grow except under special and controlled conditions; these can typically be categorized as either surfactant-assisted synthesis or the assembly of simpler 1D NCs [6]. It has been widely reported that surface modification of TiO_2_ with noble metal nanoparticles (NPs), such as from Pt, Pd, Cu, Ag, and Au, could improve its photocatalytic activity [7,8,9,10]. Special attention has been devoted to Ag-modified TiO_2_ because of the capability of Ag NPs to prevent the recombination of electron/hole (e−/h+) pairs due to the formation of the Schottky barrier at the interfaces with TiO_2_ and to act as a trapping center for photoexcited electrons. Thus, the role of Ag NPs as electron scavengers is vital in suppressing (e−/h+) recombination and achieving successful improvement of Ag-TiO_2_ photocatalytic performance. A review from 2021 with the title “Recent progress on Ag/TiO_2_ photocatalysts: photocatalytic and bactericidal behaviors” reported more than 500 papers focused on the physicochemical properties of Ag-TiO_2_ nanoscale materials [11]. Several distinct routes have been applied for Ag-TiO_2_ synthesis: for instance, sol-gel [12,13], co-precipitation [14], and hydrothermal methods [15,16], leading to bulk polycrystalline powders, as well as deposition/reduction techniques [17,18], resulting in thin layered films. Each preparation technique has been used to obtain an efficient, stable, and scalable photocatalyst. However, there are only a few reports focusing on the 2D TiO_2_ nanosheets decorated with Ag NPs obtained from cryo-lyophilized precursors [19,20]. The cryo-lyophilization technique (CLT) was first developed from wet shaping techniques, in line with the global scientific and technological interest in developing cleaner and more efficient “green” methods of synthesis. The CLT uses little other than water, which is an environmentally friendly, nontoxic dispersing medium and costs very little as a tool, and represents huge savings compared to the sol-gel method [7]. To date, few efforts have been directed towards the growth of metal-decorated TiO_2_ nanosheets using CTL, which can be attributed to the preferential application of this technique in bio/pharmaceutical and food industries instead of in material sciences [21,22]. Herein, the CLT and laser ablation deposition techniques [23,24] were used to produce Ag_TiO_2_ nanosheets and Ag_TiO_2_ nanostructured thin films (NSTFs), respectively. The synthesis efficiency was compared in terms of morphology and photocatalytic activity. The CLT of a frozen aqueous mixture of TiOSO_4_ and Ag(NO)_3_ led to amorphous peroxo-polytitanic Ti^(IV)^ acid (PPTA) precursor, which transformed into an Ag^(0)^_TiO_2_ (anatase) with a stacked nanosheet morphology after annealing at temperatures between 500 and 950 °C. In addition, the CLT conditions (freezing time, freezing temperature, and freezing rate) resulted in differently shaped nanocavities in the Ti-O layers due to the removal of ice/water through sublimation. Likewise, the laser ablation techniques yielded TiO_2_ (anatase) NSTF decorated with Ag^(0)^ NPs after 1 h heat treatment of the as-deposited film at a temperature of up to 500 °C. Additionally, the photocatalytic activity of Ag^(0)^_TiO_2_ nanosheets was tested under UV light in terms of its ability to degrade 4-chlorophenol (4-CP) and perform total organic carbon (TOC) removal. The photoelectrochemical (PEC) properties of Ag^(0)^_TiO_2_ NSTFs were examined under visible light irradiation. The photocatalytic activity of Ag^(0)^_TiO_2_ materials, even prepared via completely different techniques, was found to be correlated with the multiply twinned structure formed in Ag^(0)^ NPs with diameters in the 10–20 nm range. We evaluated the catalytic activities of the Ag^(0)^_TiO_2_ materials against the commercial TiO_2_ and standard TiO_2__P25 catalysts.

## 2. Experimental Procedure 

### 2.1. Materials 

Titanium (IV) oxysulfate (titanyl sulfate, TiOSO4, Sigma-Aldrich spol. s.r.o., Prague, Czech Republic) was used as a TiO_2_ precursor. Silver nitrate, AgNO_3_ ≥ 99%, Sigma-Aldrich spol. s.r.o., Prague, Czech Republic, served as the Ag dopant source. During material synthesis, an aqueous solution of ammonia (NH_3_, solution p.a., 25–29%, Fisher Scientific, spol. s.r.o., Pardubice, Czech Republic) was used for precipitation of the precursor and hydrogen peroxide (H_2_O_2_, solution p.a., 30%, Fisher Scientific, spol. s.r.o., Pardubice, Czech Republic) for pH reduction. 4-chlorophenol (4-CP, purity ≥ 99.9 %, Sigma-Aldrich, Prague, Czech Republic) and AEROXIDE TiO_2_-P25 (Evonik, Prague, Czech Republic) were used for the photocatalytic experiments.

### 2.2. Sample Preparation 

#### 2.2.1. Ag_TiO_2_ Nanosheets

In a typical experiment (see Figure 1 and Appendix A), 4.80 g of titanyl sulfate (TiOSO_4_) was dissolved in 150 mL of distilled water at 35 °C (step 1). Then, 0.05 g of AgNO_3_ solution was added, yielding a calculated value of 3 wt. % of silver to give a colorless solution. This solution was cooled for approximately 1 h in the freezer, and thereafter the precipitation was carried out at ~0 °C in aqueous ammonia until the pH reached 8. The precipitate was filtered and washed several times to remove sulfate anions formed during the reaction. The precipitate as obtained was transferred into a beaker and resuspended into 350 mL of distilled water. The pH of the resulting suspension was reduced by adding 20 mL of 30% hydrogen peroxide (H_2_O_2_), and it was stirred at ambient temperature until the solution turned from turbid yellowish to transparent yellow. The PPTA solution was added dropwise into Petri dishes immersed in liquid nitrogen (LN_2_) (step 5). Frozen droplets were immediately lyophilized at 10 mTorr, at −54 °C for 48 h by using VirTis Benchtop K, Core Palmer UK lyophilizer (step 6). After lyophilization, a yellowish lyophilized precursor (lyophilized foam-like cake) was isolated (step 7) [25,26,27,28]. The lyophilized precursor labeled Ti_Ag_LYO was further heat-treated under air at 500, 650, 800 and 950 °C for 1 h in each case (with rate 3 °C/min), and four new samples denoted as Ag_TiO_LYO/500, Ag_TiO_LYO/650, Ag_TiO_LYO/800 and Ag_TiO_LYO/950 were isolated (see Appendix A/Figure 1).

#### 2.2.2. Ag_TiO_2_ Nanostructured Thin Films (NSTFs)

The samples were prepared by ArF laser ablation (wavelength 193 nm, 100 mJ/pulse) of TiO_2_ and elemental Ag targets. The ablation of the sintered TiO_2_ was carried out in a turbomolecular vacuum (10^−3^ Pa) using a focused laser beam (9 min, 10 Hz) to prepare a thin film. Subsequently, the thin film was covered by Ag NPs, prepared by laser ablation under argon (4 Pa). The deposits were grown on NaCl, quartz and Cu substrates, and FTO glass. The samples were annealed at 500 °C/1 h under air to crystallize the TiO_2_ thin film and to form Ag NPs on the surface. The two samples (1) labeled Ag_TiO_2__AP, as prepared before annealing, and (2) annealed at 500 °C—Labeled Ag_TiO_2__500 were the subjects for further characterization.

### 2.3. Characterization Methods

Powder diffraction patterns of samples Ag_TiO_LYO/500, Ag_TiO_LYO/650, Ag_TiO_LYO/800, and Ag_TiO_LYO/950 were collected using a PANalytical X’ıPertPRO diffractometer equipped with a conventional X-ray tube (CuK _40 kV, 30 mA, line focus) in transmission mode. An elliptic focusing mirror, a divergence slit of 0.5°, an anti-scatter slit of 0.5°, and a Sollerslit of 0.02 rad were used in the primary beam. A fast-linear position-sensitive detector PIXcel with an anti-scatter shield and a Soller slit of 0.02 rad was used in the diffracted beam. All patterns were collected in the range of 18–88 2theta with a step size of 0.013 and 400 s/step, producing a scan of about 2.5 h. Qualitative analysis was performed with the HighScorePlus software package (PANalytical, Almelo, The Netherlands, version 3.0e) and the DiffracPlus software package (Bruker AXS, Karlsruhe, Germany, version 8.0) [29]. For quantitative phase analysis, we used DiffracPlus Topas (Bruker AXS, Karlsruhe, Germany, version 4.2) [30]. The estimation of the size of crystallites was performed based on the Scherrer formula [31] (Equation (1)).
(1)$$D=\frac{K\lambda }{\beta \cos\theta }$$
where *K* stands for Scherrer’s constant, *λ* corresponds to the X-ray wavelength irradiation, *β* is the half-width of the diffraction peak and θ corresponds to the scattering angle. The model was derived from anatase TiO_2_ structure with Ti atoms replaced by 1 at. % Ag.

Transmission electron microscopy (TEM) of Ag_TiO_2_ nanosheets was carried out on an FEI Tecnai TF20 X-twin microscope operated at 200 kV (FEG, 1.9Å point resolution) equipped with an EDAX Energy Dispersive X-ray (EDX) detector (FEI company, Hillsboro, OR, USA). The microscope was used in scanning mode (STEM) with High-Angle Annular Dark Field Detector (HAADF). TEM images were recorded on a Gatan CCD camera with a resolution of 2048 × 2048 pixels using the Digital Micrograph software package. Selected area electron diffraction (SAED) patterns were evaluated using the Process Diffraction software package [32]. For the TEM, the thin film deposits were prepared on NaCl substrates (at the same experiment with the other substrates) to facilitate the sample preparation which was simply done by dissolving the substrate and placing the thin film deposit on a copper grid.

High-angle annular dark-field scanning transmission electron microscopy (HAADF-STEM) images of Ag_TiO_2_ NSTFs were acquired using cold-field-emission double aberration-corrected JEOL JEM-ARM200CF microscope operated at 200 kV. The inner-collection semi-angle of the HAADF detector was set to 90 mrad, and the probe convergence semi-angle was 22 mrad. X-ray energy dispersive spectroscopy (EDS) mappings were acquired using 0.98-steradian solid-angle windowless silicon drift-detector JED-2300 (JEOL, Tokyo, Japan) mounted in the STEM instrument. For the atomic-resolution EDS, the probe current was set to 200 pA, 0.1–0.2 msec pixel dwell time. Wiener filtering and sample drift correction was applied, 5–6 sweeps were accumulated for the atomic-resolution EDS images. 

The surface composition of the samples, chemical states, and electronic states of the elements was inspected by XPS using Kratos ESCA 3400 furnished with a polychromatic Mg X-ray source of Mg Kα radiation (energy: 1253.4 eV). The base pressure was kept at 5.0 × 10^−7^ Pa. The spectra were fitted using a Gaussian–Lorentzian line shape, Shirley background subtraction, and a damped non-linear least square procedure. Spectra were taken over Ti 2p, O 1 s, C 1 s, and Ag 3d regions. The samples were sputtered with Ar^+^ ions at 1 kV with a current of 10 µA for 60 s to remove superficial layers. Spectra were calibrated to C 1 s line centered at 284.8 eV.

The surface nanostructure of the synthesized Ag_TiO_2_ nanosheets was analyzed by atomic force microscopy (AFM) using NTEGRA AFM system (NT-MDT). The ex-situ AFM measurements were carried out in tapping mode under ambient conditions (room temperature, air environment, normal atmospheric pressure). Atomic Force Microscopy (AFM) measurements of Ag_TiO_2_ NSTFs were carried out at room temperature on an ambient AFM (Bruker, Dimension Icon, Ettlingen, Germany) in Peak Force Tapping mode with ScanAsyst Air tips (Bruker; k = 0.4 N/m; nominal tip radius 2 nm). The measured topographies had a resolution of 512 × 512 points^2^.

UV-Vis spectroscopy was carried out using a Shimadzu spectrophotometer UV 1800 (Kyoto, Japan). The sample was dispersed in distilled water in a quartz cell (10 mm optical path) in an ultrasound bath for 2 min. The spectra were recorded in the range 190–800 nm. Diffusion reflectance spectra of the powders were measured after fixing the samples on double-sided carbon tape. The thin films were measured as prepared and annealed on quartz substrates.

Pin-hole Small-Angle Neutron Scattering (SANS) measurements of the samples Ag_TiO_LYO/500, Ag_TiO_LYO/650, Ag_TiO_LYO/800 and Ag_TiO_LYO/950 were conducted using a conventional SANS V4 instrument (Helmholtz-Zentrum Berlin für Materialien und Energie. (2016). V4: The Small Angle Scattering Instrument (SANS) at BER II [33] located at BER-II reactor of Helmholtz-Zentrum Berlin). Neutron scattering curves were measured using a collimated neutron beam with a wavelength of 0.5 nm (±10%) at distances from the sample to the detector of 15.78 m, 6.8 m, and 2 m. The data were treated using a standard procedure using “empty cell” and “cadmium background” measurements. The samples were installed for the SANS measurements in quartz cells with a flight path of 1 mm. 

### 2.4. Photocatalytic Degradation of 4-CP

The prepared photocatalysts (0.15 g L^−1^) were added in an aqueous 4-chlorophenol solution (4-CP, 1.0 × 10^−4^ mol L^−1^). The reactant mixtures were irradiated under identical conditions in a photoreactor equipped with 10 lamps (Sylvania Blacklight 8 watt, *λ*_max_ = 368 nm, intensity 6.24 mW cm^−2^). The volume of the reactant mixture was 175 mL. The photocatalytic activity of the samples was monitored by measuring the concentration of 4-CP in water (HPLC, Agilent Technologies 1200 Series, column LiChrospher RP-18, 5 μm, mobile phase mixture of methanol and water, UV-Vis absorption detection) as well as the total organic carbon (TOC-LCPH, Shimadzu, Kyoto, Japan). The instrumental method employs the incineration of the sample at 680 °C, resulting in the formation of CO_2_. Inorganic carbon presented as carbonate is transformed with 0.1% HCl to CO_2_. Formed CO_2_ was detected by implementing IR absorption spectroscopy. Water samples (20 mL) were taken periodically at 60, 120, and 240 min of irradiation.

The reaction rate of 4-CP reduction was fitted to the pseudo-first-order kinetic model (Equation (2)):(2)$$\ln\;(\frac{C}{C_{0}})=-kt\;$$
where *C* and *C*_0_ are the summed 4-CP concentrations at the time (*t*) and *t* = 0, respectively, and *k* is the pseudo-first-order kinetic rate constant [34]. The method of UV-Vis diffuse reflectance spectroscopy was employed to estimate band-gap energies of the prepared Ag decorated TiO_2_ nanosheets. Diffuse reflectance UV-Vis spectra were recorded in the diffuse reflectance mode (*R*) and transformed to a magnitude proportional to the extinction coefficient (*K*) through the Kubelka–Munk function. A PerkinElmer Lambda 35 spectrometer equipped with a Labsphere RSA-PE-20 integration sphere using BaSO_4_ was used as a standard. The band-gap energy *E*_bg_ [35] was calculated by the extrapolation of the linear part according to Equation (3)
*λ*_bg_ = 1240/*E*_bg_ (eV) (3)

### 2.5. Photoelectrochemical Measurements

The photocatalytic activity for water splitting was tested by cyclic voltammetry on samples deposited on the FTO glass substrates. Cyclic voltammetry (CV) was carried out under irradiation by a visible light sun simulator (100 W, Oriel LCS 100). CV was performed using a scan rate of 20 mV/s at potentials −0.5 and +1.5 V with Pt counter electrode, Ag/AgCl reference electrode in 0.5 M H_2_SO_4_ as an electrolyte. To demonstrate water splitting performance, the samples were studied by linear sweep voltammetry (LSV) between −0.2 and +1.2 V by applying 10 s light/10 s dark cycles at 1 mV/s scan rate.

## 3. Results and Discussions

### 3.1. Microstructure and Surface Characterization of Ag_TiO_2_ Nanosheets

X-ray powder diffraction analysis and Rietveld refinement were used for characterization of the microstructure and evaluation of TiO_2_ phase transition. The characteristic parts of XRD patterns of precursor Ag_TiO_LYO and post-synthesis annealing products Ag_TiO_LYO/500, Ag_TiO_LYO/650, Ag_TiO_LYO/800, and Ag_TiO_LYO/950 are shown in Appendix A. XRD pattern of the lyophilized precursor Ag_TiO_LYO indicated amorphous material. The XRD patterns of samples Ag_TiO_LYO/500, Ag_TiO_LYO/650, and Ag_TiO_LYO/800 revealed anatase phase only; the positions of diffraction peaks and distribution of intensities correspond to anatase TiO_2_ (ICDD PDF No 21-1272) [28]. Increasing the temperature resulted in more refined reflection peaks i.e., reflections became narrower as the crystallites grew from 35.37 nm at 500 °C to 78.41 nm at 800 °C (Table 1) [36]. It is evident that the Ag_TiO_LYO/950 sample already contains 66.53% of rutile (JCPDS PDF No. 21-1276) [28] and a small fraction (0.7 %) of cubic (Ag^0^) with JCPDS PDF No. 04-783 [37]. The difference in the crystal structure in our 2D TiO_2_ materials with different anatase/rutile ratios is due to thermal treatment and spontaneous transformation from metastable anatase to stable rutile [38]. XRD results confirmed that the metallic (Ag^0^) form of silver stabilized the anatase phase up to 800 °C and retarded phase transformation in comparison with pristine TiO_2_ nanosheets prepared by using the same method [39,40].

It was found that the lattice parameter a was not changed, whereas the lattice parameter *c* increased linearly with an increase in temperature. The visible extension along the *c*-axis and the resulting lattice expansion could be dependent on the nanoparticle size and thermal treatment. A considerable increase in the anatase crystallite size was observed after annealing at an overall temperature interval of 500–950 °C. The crystallite size of sample Ag_TiO_LYO/950, as determined from the XRD, was 238.7 (anatase) vs. 251.1 (rutile), since the transformation of anatase to rutile is usually accompanied by crystal growth through the coalescence process [41]. 

The presence of metallic (Ag^0^) silver was evidenced by the high-resolution XPS spectra, as depicted in Figure 1. 

Two well-defined peaks centered at 368.0 and 374.0 V were observed in the Ag 3d region of all samples. The bands were narrow (FWHM = 1.2 eV), which means that silver was present in one chemical state. Both bands had an asymmetric shape, which is typical for the metallic (Ag^0^) form of silver. Loss features usually observed due to the higher binding energy side of each spin-orbit component were not visible due to low surface concentrations of elemental Ag^(0)^ silver. Ag-O bonds, which could be expected as a result of exposure of the samples to ambient air, were not visible in either the prepared or sputtered samples. Ag-O bands had binding energy about 0.5 eV lower than elemental Ag^(0)^, and therefore they were observed as a broadened feature on the lower-binding-energy side of the Ag 3d bands.

The Ti 2p region showed broad bands and a Ti 2p_3/2_ band separated into three contributions, with the highest content typical for Ti^4+^ in TiO_2_ (458.8 eV). Lower energy bands were ascribed to Ti^3+^ valence states in titanium suboxides. Titanium represents an element with a constant surface concentration, and the intensity of the Ag 3d peaks shows that the silver concentration changed significantly with annealing temperature. The surface concentration of silver was lowest at 500 °C (1.7% relative to Ti), while heating to 650 and 800 °C led to an increase in Ag^(0)^ surface concentration (1.8 and 6.9%, respectively, to Ti). Heating to 950 °C again resulted in lower Ag^(0)^ surface concentration, probably due to the evaporation of elemental silver at the highest temperature. The O 1 s spectra of the sample were deconvoluted into three peaks. The most intense one, centered at 530.2 eV, is typical for metal oxides. The higher binding energy contributions were assigned to hydroxide and carbonate (531.6 eV) and adsorbed water (533.2 eV). 

The morphology of annealed precursor Ag_TiO_LYO was observed by SEM, and the results for Ag_TiO_LYO/500, Ag_TiO_LYO/650, Ag_TiO_LYO/800, and Ag_TiO_LYO/950 are presented in Figure 2a–d. The annealing process led to the formation of network-like joined nanosheets, which are distinguishable by their shape and size [36]. SEM observation shows that crystallization took place only in the 2D direction, and 2D nanosheets with different degrees of crystallinity were acquired. Even at 950 °C, when the anatase–rutile transformation took place, the 2D nanosheet morphology did not collapse, and its morphology was preserved (Figure 2d).

In-depth morphology evaluation was performed by HAADF-STEM monitoring. Figure 3a verifies the nanosheet morphology of the precursor Ag_TiO_LYO.

The Z-contrast HAADF-STEM image (Figure 3b) and STEM-EDS mapping (Figure 3c) show that Ag is segregated on the edge of the nanosheet. After annealing at 500 and 650 °C for 1 h, it can be seen that the microstructure and roughness of the Ag_TiO_LYO were significantly changed. Low-magnification TEM of the Ag_TiO_LYO/500 sample is presented in Figure 4a. The high-resolution images in Figure 4b–c, taken from the yellow boxed area in Figure 4a, show spheroidal-shaped NPs, with a diameter around 20 nm, attached to the surface of nanosheets. HRTEM analysis (from the yellow marked area in Figure 4c) suggested a single Ag^(0)^NP with the cubic *Fm*-3*m* space group. The calculated *d*-spacing of 0.235 nm corresponds to the (111) plane of Ag^(0)^ NP (JCPDS PDF No. 04-783) when oriented in the [103] zone axis (Figure 4c_1_). 

It is worth noting that the cubic Ag^(0)^ NP structure was preserved; indeed, there was an evolution in particle shape from a spheroid to a spherical morphology during annealing at 650 °C. Ag^(0)^ NPs oriented along their five-fold decahedral symmetry with five merged subunits (Figure 4e) were tightly attached on the surface of well-crystallized TiO_2_ (Figure 4e). Indeed, the measured d_(101)_ = 0.358 nm was attributed to tetragonal anatase with space group *I*41/*amd* (see Figure 4f) and PDF JCPDS 21-1272. Figure 4f_1_ shows the HRTEM image obtained by DigitalMicrograph analysis (provided from the area highlighted in yellow in Figure 4f), suggesting a lattice spacing of 0.202 nm, corresponding to the (101) lattice planes of cubic Ag^(0)^. Additionally, the uniform d-spacing along different 〈100〉 directions is consistent with the face-centered cubic (fcc) phase of metallic Ag^(0)^ NPs, confirming that our observations are in line with the statement that the fivefold twinning phenomenon is very common for metal Ag [42,43,44]. Additionally, the HRTEM findings corroborated the observed XRD patterns of Ag_TiO_LYO/500 and Ag_TiO_LYO/650 (see Table 1 and Appendix A). 

Figure 5 shows the aberration-corrected bright-field (BF) and HAADF STEM micrographs of Ag_TiO_LYO/800 acquired at different magnifications. Two regions can be distinguished in the HAADF-STEM images in Figure 5a–b: a matrix with lower intensity, and well-crystallized spherical NPs that present higher intensity. This contrast is associated with the atomic weight dependence of the constituted elements. Because the brightness of an individual atomic column in a HAADF-STEM image (Figure 5b) is approximately proportional to the square of average atomic number, the contrast of Ag (*Z* = 47) appears brighter than that of Ti (Z = 22) in the HAADF electron scattering regime. Therefore, the spherical NPs that were brighter, in contrast, indicated enrichment of Ag^(0)^. As a result of increasing temperature, the size of Ag^(0)^ NPs changed, appearing as smaller, evenly distributed Ag^(0)^ NPs on the surface of the matrix (TiO_2_ nanosheets), coexisting with a small number of larger Ag^(0)^NPs segregated at the grain boundaries between two adjacent (TiO_2_) matrix grains. Indeed, the zoomed BF STEM image in Figure 5c taken from the yellow boxed area in Figure 5a comprises an atomic-resolution image of the matrix, verifying the well-crystallized nanograins (NGs) with an interlayer spacing of 0.35 nm between the (101) planes proposed for anatase TiO_2_. In Figure 5d, we present the BF STEM view of a ⟨110⟩-oriented Ag^(0)^ NP with the characteristic five-fold twinned nanostructure. Five subunits, labeled T1−T5, are joined together, sharing their {111} planes. In regular fcc metal Ag^(0)^ with five ideal single-crystalline grains joined in a fivefold twinned nanostructure without distortion, the angle between two (111) faces is known to be 70.53° [45]. However, by applying CrytalMaker software [46], we found that the angle along the [110] zone axis was 74.5. Therefore, a significant angular mismatch between T1−T5 occurred in the Ag_TiO_LYO/800 sample upon annealing at 800 °C. Such angle subtending should be compensated by simultaneously adjusting the bond lengths and the introduction of lattice defects such as dislocations and stacking faults (SF) in Ag^(0)^NPs [43]. If we accepted the model that any T1−T5subunits can be regarded as a single sub-crystal merged in a pentagonal bipyramidal geometry, we could suggest that lattice strain and distortion may be generated from the matching adjacent {111} twin boundaries (TBs) upon annealing. From Figure 5c, it appears that the atomic arrangement of the TiO_2_ grains with d_(101)_ = 0.35 nm is preserved, whereas the atomic arrangement of Ag^(0)^ NPs at the surface is highly distorted (Figure 5f).

Detailed examination of Ag–Ag bond lengths by CrystalMaker demonstrates the presence of distortion within the exanimated pentagonal bipyramidal core (highlighted by the red circle in Figure 5f). It was observed that Ag–Ag bonds are longer (0.291 nm) than standard 0.287 nm for bulk Ag^(0)^. The analysis suggested that the geometric distortion in the decahedral core was compensated by concomitant distortions along with the fivefold T1–T5 axis. Twinning by a mirror plane, where the twinned Ag^(0)^ lattice is obtained by a homogeneous simple shear of the original cubic lattice, is well visible in subunit T2 (Figure 5f). The two mirror regions are referred to as twin domains 1 and 2, respectively. Therefore, the Ag^(0)^ NPs showed a strong tendency to form multiply twinned face-centered cubic superlattices with decahedral symmetry when their size was reduced to 10 nm (Figure 5f) [47,48].

Figure 5f also includes subunit T5, where the SF implies the dislocation lines, observed as a result of TB interaction. The HAADF-STEM examination confirmed that higher temperature was able to facilitate the progressive coalescence of two Ag^(0)^NPs into one larger Ag^(0)^NP with a fivefold twinned structure [49]. Additionally, we could suggest that the perfect anatase TiO_2_(101) may provide nucleation sites for the growth of Ag^(0)^NPs with a pentagonal bipyramidal structure (Figure 5c), which, unlike Ag^(0)^NPs with a planar (flat) geometry in Ag_TiO_LYO)/500 (Figure 5a–c), could support the improvement of the properties of Ag_TiO_LYO)/800 as a photocatalyst [50]. The STEM image of Ag_TiO_LYO/950 with corresponding SAED (Appendix A) confirms single rutile nanocrystals with a size larger than 200 nm, which is in line with the XRD results (Table 1). 

Appendix A presents a visualization of the surface topology of the annealed Ag_TiO_LYO/800 material using the AFM technique. Taking into account the geometrical similarities of the NPs, it is reasonable to also expect their identical chemical origin. Careful analysis of the nanostructure allowed us to distinguish the formation of the 2D nanosheets between the NPs, suggesting the formation of a new phase, which is likely to be different from the phase of the NPs (see Appendix A). Therefore, the performed analysis evidences the formation of two different phases. Appendix A presents the quantitive analysis of Ag_TiO_LYO/800 using surface profile plots. An NP height of 100 nm was estimated.

SANS analysis was further performed in order to examine the NP size of the Ag_TiO_LYO/500, Ag_TiO_LYO/650, Ag_TiO_LYO/800, and Ag_TiO_LYO/950 samples. The obtained results show a prominent local increase in intensity at Q~0.2 ÷ 0.35 nm^−1^ (see Figure 6). 

The data were fitted using a model of spherical particles with a log-normal distribution of radius as follows:(4)$$N\left(r\right)=\frac{1}{\sigma \sqrt{2\pi }}exp\left(-\frac{1}{2}{\left(\frac{r-r_{0}}{\sigma }\right)}^{2}\right)$$

The fitted size parameters were *r*_0_ = 2.3 nm and *r*_0_ = 1.92 nm for the Ag_TiO_LYO/500 and Ag_TiO_LYO/650 samples, respectively. The calculated size distributions were quite wide (*σ* = 0.7 ÷ 0.8), showing the high polydispersity of the scattering objects. In contrast to this outcome, the SANS data for the samples treated at higher temperatures—Ag_TiO_LYO/800 and Ag_TiO_LYO/950—did not show any deviation from the Porod scattering [39] (Figure 6b), and nanosized objects were not detected by SANS. This observation is in agreement with the XPS experiment, confirming that heating to 800 °C leads to increasing of Ag surface concentration with 6.9% suggesting that Ag NPs, with their increased surface free energy during annealing, can subsequently promote sintering and agglomeration of Ag^(0)^ on the surface of 2D TiO_2_ nanosheets [41]. 

Our detailed HAADF-STEM analysis further demonstrated that the geometric non-ideal decahedral Ag^(0)^ core and defected T1-T5 sub-NCs progressively coalesce into one larger Ag^(0)^ NP with a fivefold twinned structure in order to minimize the total potential energy of Ag^(0)^ NPs. 

The (UV-Vis) light absorption spectra (Figure 7) revealed that the loading of Ag could enhance the light absorption of the TiO_2_ nanosheets. The UV-Vis spectra of the Ag_TiO_LYO/500, Ag_TiO_LYO/650, Ag_TiO_LYO/800, and Ag_TiO_LYO/950 samples were measured in distilled water, having no absorption. 

All samples were highly homogeneous, and repeated measurements afforded reproducible spectra. Samples annealed at 500 and 650 °C show the most intense bands in the UV region, with maxima at 296 nm and 322 nm, respectively. Both samples are grey. The sample annealed at 800 °C with dark grey color shows a broad absorption band at 350 nm, with pronounced absorption in the visible region. The last sample, annealed at the highest temperature, exhibited absorption only in the visible region beginning at 395 nm. The differences observed are caused by structural changes in TiO_2_ and both structural and concentration changes in metallic Ag^(0)^ NPs. Diffuse reflectance spectra of the samples (Figure 7) were measured against PTFE, showing a reflectance greater than 99% throughout the measured range (220–700 nm). The spectra of the samples evidenced a major drop in the reflection in the interval 350–420 nm, which is connected to structural changes due to thermal treatment. 

The diffuse reflectance spectra of samples annealed at 650–900 °C consist of a flat, highly reflective region at long wavelengths that abruptly transforms into a steeply falling reflectance edge at shorter wavelengths. On the other hand, the sample annealed at 500 °C exhibited reflection spectra with two maxima, due to its low crystallinity, numerous defects, and the presence of carbon impurities. The diffuse reflectance spectra were transformed to a Tauc plot (Figure 7, inset), proving an indirect allowed transition in the prepared materials annealed at temperatures 650 °C and higher. Samples Ag_TiO_LYO/650 and 800 possessed band gaps at 3.23 eV, which is typical for an anatase structure, but the higher-temperature sample (Ag_TiO_LYO/800) had a much higher silver surface concentration, which explains it’s having the highest photocatalytic activity. A bandgap of about 3.01 eV was observed in the Ag_TiO_LYO/950 sample due to transformation to rutile, known for lower photocatalytic activity. 

### 3.2. Photocatalytic Decomposition of 4-CP and TOC

HPLC and TOC measurements were applied to obtain a better understanding of the photocatalytic performance of 4-CP under UV irradiation. We observed that the highest 4-CP removal was obtained in the presence of the Ag_TiO_LYO/800 photocatalysts, followed by Ag_TiO_LYO/650 and Ag_TiO_LYO/500, and Ag_TiO_LYO/950 (Figure 8). 

The calculated degradation rate constants *k* (s^−1^) are shown in Table 1. The obtained *E*_g_ values (Appendix A) were correlated based on UV-Vis analysis (Figure 7) and the Kubelka–Munk function.

The best performance in 4-CP photocatalytic decomposition was achieved with the Ag_TiO_LYO/800 material, which had the lowest *E_b_**_g_* and redshift of optical absorption (Appendix A). The *E**_g_* value of Ag_TiO_LYO/800 was lower than that of the reference anatase sample (*E**_g_* = 3.24 eV), and pristine 2D TiO_2_ nanosheets obtained by the same method were reported in our previous article [39]. Briefly, the *E**_g_* values for the pristine samples were estimated as TiO_LYO (3.19 eV), TiO_LYO/500 (3.24 eV), TiO_LYO/650 (3.24 eV), TiO_LYO/800 (3.24 eV), and TiO_LYO/950 (3.23 eV). The higher photocatalytic efficiency of the Ag_TiO_LYO/800 catalyst can be explained by the even distribution of Ag^(0)^ on the TiO_2_ (anatase) nanosheet surface, with an average size of 5–10 nm, as well as fivefold twinned Ag^(0)^, which has a clear positive effect on the photocatalytic activity under UV light irradiation [51,52]. Additionally, at a temperature of 800 °C, the Ag^(0)^ NPs stabilized the anatase structure, decreasing the size of the crystallites compared to pristine TiO_2_ nanosheets. Further annealing at 950 °C leads to a decrease in photocatalytic efficiency of the Ag_TiO_LYO/950 material under UV light. The differences in the particle shape affect the redistribution of the Ag^(0)^ NPs on the Ag_TiO_LYO/950 surface, resulting in its lower photo efficiency. Annealing at a temperature of 950 °C resulted in a full conversion from anatase to rutile and a change in all microstructural characteristics and photocatalytic activity. Herein, we used the TOC removal as an inexpensive test to evaluate the Ag_TiO_LYO/500, Ag_TiO_LYO/650, Ag_TiO_LYO/800, and Ag_TiO_LYO/950 materials. The TOC removal over time is illustrated in Figure 9.

Degradation of 4-CP in the presence of Ag_TiO_LYO/500, Ag_TiO_LYO/650, Ag_TiO_LYO/800, and Ag_TiO_LYO/950 photocatalysts is rapid in the first 120 min of treatment but decreases thereafter. TOC removal in the range of 2.7 to 12% was achieved in the measured samples after 240 min of photocatalytic treatment. The highest degree of mineralization (12% TOC removal) was obtained in the presence of the Ag_TiO_LYO/650 photocatalyst. 

### 3.3. Structural Analysis and Surface Characterization of Ag_TiO_2__AP and Ag_TiO_2__500 NSTFs

Nanostructured thin films (NSTF) prepared by laser ablation have extremely high adhesion to all substrates. The as-prepared (Ag_TiO_2__AP) deposit was greyish, but after annealing at 500 °C (Ag_TiO_2__500), it changed color and the samples became pink with a metallic luster. 

To evaluate the microstructure and phase composition of Ag_TiO_2__AP and Ag_TiO_2__500 NSTFs, the samples were studied using various TEM techniques: imaging (including HRTEM), SAED, STEM-HAADF, and EDX mapping. In addition, the surface and thickness of the deposits were investigated using AFM.

Figure 10 shows TEM observations of Ag _TiO_2__AP and Ag_TiO_2__500 NSTFs acquired at low magnification, displaying the different morphologies of the nanoparticles on the surfaces of the NSFTs. Ag_TiO_2__AP contains nanoparticles with irregular shapes with sizes within a wide range from 10 to 100 nm. In contrast, the annealed sample contained spherical nanoparticles with sizes within the narrower range of 20–40 nm, homogeneously distributed on the TiO_2_ surface. The corresponding SAED diffraction patterns were used to determine the presence of Ag and TiO_2_ and their structural form. The SAED examination of Ag _TiO_2__AP (Figure 10b) confirmed the mixture of two phases: a broad halo coexisting with several sharp concentric diffraction rings. The disordered phase belonged to the amorphous TiO_2_ (matrix), whereas the diffraction rings corresponded to polycrystalline Ag^(0)^ with a cubic structure in the *Fm*-3*m* space group and JCPDS PDF No. 87-0718 (Figure 10c). The SAED pattern of the annealed Ag_TiO_2__500 sample (Figure 10e) also confirmed the mixture of two phases; however, these were slightly different from Ag _TiO_2__AP. The diffraction rings corresponding to polycrystalline Ag (Figure 10f) were still present, but the broad halo of the amorphous TiO_2_ was replaced by intense spots corresponding to the single-crystal orientation of anatase viewed down [110]. In addition, the TiO_2_ film had crystallized into large single-crystal anatase grains that were slightly bent, as evidenced by the bending contours in Figure 10d. Therefore, it can be inferred that microstructural evolution took place during the annealing process; the Ag_TiO_2__500 NSTFs consisted of well-crystallized anatase TiO_2_ (after the transition from amorphous TiO_2_) and uniformly dispersed Ag^(0)^ NPs. 

The TEM results were further corroborated by the STEM/HAADF images and STEM/EDX mapping of Ag_TiO_2__AP and Ag_TiO_2__500 NSTFs (Figure 11). The white contrast in the HAADF images indicates the heavy Ag on the light support of TiO_2_, displayed in dark grey or black. Moreover, the EDX maps show the Ag, Ti, and O elemental distribution over the film in detail. It can be seen from the maps of Ti and O that both elements become interwoven and distributed all over the film, whereas the Ag map suggests that Ag is not interconnected with either Ti or O. This observation indicates that Ag forms a separate phase, which was also evidenced by the SAED observations (Figure 10).

A detailed understanding of the Ag_TiO_2__500 NSTF growth and morphology was further investigated by HRTEM analysis (Figure 12). The high-magnification HRTEM image in Figure 12b corresponds to Ag _TiO_2__AP, in which Ag single-crystal NPs with quasi-spherical shape and an average particle size close to 20 nm can be easily recognized. Figure 12d–f shows the results for Ag_TiO_2__500. Upon annealing at 500 °C, the original Ag NPs were completely rearranged into spherical NPs. In addition, these spherical NPs were very often composed of the intergrowth of multiple individuals with twinning planes {111}. The most common was the well-known pentagonal intergrowth, with a common zone axis [110] for all the individuals [42,43,44]. The formation of such NPs could be the result of the “dissolution–recrystallization” process of Ag agglomerates formed by the gathering of smaller quasi-spherical Ag NPs upon annealing. This is further supported by the determination of particle size redistribution by the ImageJ software (Appendix A). The obtained results indicate that annealing at 500 °C yielded significantly different particle size distributions for Ag_TiO_2__500 than for Ag_TiO_2__AP. These different particle sizes may represent the mass balance between the constituent element (crystallization of anatase from amorphous TiO_2_), and the formation of smaller, more stable spherical Ag NPs. Therefore, it can be concluded that the annealing process can modify the entire morphology of Ag _TiO_2__AP.

The surface topography of Ag_TiO_2__AP and Ag_TiO_2__500 NSTFs was studied by AFM (Figure 13). It was observed that continuous and uniform films were produced without a visible change in topography. From the surface profile plots measured along the horizontal lines in the AFM images, it can be seen that annealing led to an increase in roughness and to a small increase in thickness from 40 to 43 nm in Ag_TiO_2__500 (Figure 13c,f). The different color of the NPs reflects their different height, suggesting the formation of agglomerates upon annealing. Analysis of the magnified images reveals the size distribution of the NPs, with a lateral size of 18–19 nm for the Ag_TiO_2__AP NSTF. Larger agglomerates with sizes of 26–27 nm appeared on the Ag_TiO_2__500 NSTFs surface. The thickness was measured at the sharp edge of the film, which was produced by applying a mask on the substrate during the deposition. The thickness was about 40 and 43 nm for Ag_TiO_2__AP and Ag_TiO_2__500 NSTFs, respectively. The higher spatial density of the agglomerations in Ag_TiO_2__500 NSTF and the shift of the size distribution towards a larger NP size were further supported by ImageJ software estimation (Appendix A). 

Ag_TiO_2__AP and Ag_TiO_2__500 NSTFs have distinct microstructures and phase compositions. Ag_TiO2_AP is composed of amorphous TiO_2_ decorated with cubic Ag particles with irregular shapes with the size in the range 10–100 nm. In contrast, Ag_TiO_2__500 is composed of crystalline TiO_2_ (anatase) thin-film decorated with spherical Ag nanoparticles with sizes in the range 20–40 nm, which are homogeneously distributed on the TiO_2_ surface (Figure 11). These particles very often exhibit multiple twinning on {111} planes (Figure 12). The film thickness was 40 nm for the Ag_TiO_2__AP sample and 43 nm for the Ag_TiO_2__500 NSTFs, as observed by AFM (Figure 13). The maximum height above the TiO_2_ film of the nanoparticles is was 16 nm for the as-prepared sample and 26 nm for the annealed sample. Thus, it can be concluded that the annealing process not only leads to phase transformation of TiO_2_ from amorphous to crystalline anatase and redistribution of Ag into spherical nanoparticles with more homogeneous distribution in size as well as spatial distribution on the surface of the thin film, but also an increase in the overall thickness of the Ag_TiO_2__500 NSTFs. 

The UV-Vis spectra of the Ag_TiO_2__AP (Figure 14a) and Ag_TiO_2__500 (Figure 14b) TNSFs were measured on the quartz substrates using the transmission technique in a range of 190–1100 nm. The undoped TiO₂ layer showed high absorption at 330 nm. The film with deposited Ag^(0)^NPs showed a shoulder at 320 nm that can be regarded as interfacial charge transfer absorption and a very broad bump above 350 nm. The broad feature had a flat maximum centered at about 850 nm, which can be attributed to localized plasmon resonance (LSPR) absorption. As the LSPR is tunable and strongly dependent on the morphology (shape and size), composition, and interaction between Ag^(0)^ NPs and the substrate, the broad feature was regarded to be a result of size distribution. Both of these absorptions could be considered to have the potential to improve the photocatalytic properties of the Ag_TiO_2_ NSTFs. Indeed, in Ag_TiO_2__AP TNSF, the TiO_2_ layer showed an intensity maximum in the UV spectrum due to partial crystallization, while Ag_TiO_2__500 TNSF demonstrated intense LSPR at about 730 nm. The absorption band was narrowed because annealed NPs are spherical and have a narrow diameter distribution, as proved by STEM and ImageJ analysis (Appendix A). The band, connected to interfacial charge transfer, observed in Ag_TiO_2__AP vanished due to the small contact area between Ag NPs and anatase TiO_2_.

Bulk stoichiometry measured by EDS revealed that the atomic ratio [Ti]/[Ag] = 12.6. The XPS spectrum showed prevailing Ag in the superficial layer of the Ag_TiO_2__AP, as demonstrated by the calculation of the elemental composition based on the Ti 2p and Ag 3d regions. The atomic ratio was [Ti]/[Ag] = 0.22 for the Ag_TiO_2__AP. Silver was present in the elemental Ag^(0)^ form (Ag 3d_5/2_ BE = 368.1 eV), but corresponding Ag 3d bands were broadened due to the nanosize effect (FWHM = 1.21 eV). The shape of the elemental Ag 3d bands was asymmetrical, as seen in Figure 15.

Photoelectrochemical measurements were conducted with deposits on FTO glass substrates. Cyclic voltammetry was studied between bias potentials of −0.5 and +1.5 V with a Pt counter electrode and an Ag/AgCl reference electrode in 0.5 M H₂SO₄ as an electrolyte. Under irradiation by visible light (100 W solar lamp), hydrogen generation on Pt and oxygen generation on working electrodes were visible, and the gas products were confirmed by mass spectrometry. Above 1.2 V, rapid degradation of the Ag_TiO_2_ samples was observed, and further LSV measurements were conducted under this potential (Figure 16) with fresh electrodes. The corresponding photocurrent–potential relationships were tested under illumination and in the dark at a low scan rate (1 mV/s). Both as-prepared and annealed samples exhibited an anode current when exposed to light compared to experiments performed in the dark. The photocurrent performance of the Ag_TiO_2__500 photoanode started at 0.23 V vs. Ag/AgCl, which corresponded to the water oxidation potential. A much higher photochemical activity was observed in the annealed Ag_TiO_2__500 TNSF sample, as was expected due to the intense (LSPR) optical absorption and better crystallinity, which are two important factors in improved PEC performance in TiO_2_ materials.

### 3.4. The Key Role of Multiply Twinned Ag^(0)^ NPs on the Photocatalytic Properties of TiO_2_ Nanosheets and TiO_2_ NSTFs

It has been well documented that when TiO_2_ particles are illuminated with UV light, electron/hole (e−/h^+^) pairs can be easily created. The separation of (e−/h^+^) is a crucial step, and the low quantum yield of any photocatalytic reactions is due to the high rate of recombination between the (e−/h^+^) pairs. In the absence of suitable (e−/h^+^) scavengers, possible recombination can occur within a few nanoseconds. A promising alternative for avoiding (e−/h^+^) recombination is surface defects, which are capable of trapping surface (e−/h^+^), and further preventing their recombination. The observed improvement in 4-CP degradation with Ag_TiO_LYO/800 and PEC activity with Ag_TiO_2__500 NSTFs was attributed to the crucial role of multiply twinned Ag^(0)^ characterized by a {111} TBs. We propose that the presence of TBs in decahedral Ag^(0)^ leads to an increase in the number of immobilized Ag atoms with dangling bonds on the anatase TiO_2_ surface [53]. It is generally known that such immobilized atoms are simultaneously highly reactive and highly unstable. After UV-light illumination, a transfer of photoexcited electrons from TiO_2_ to immobilized Ag could be achieved. The Ag could trap the electrons, and further interplay might (i) avoid (e−/h^+^) recombination, and (ii) contribute to the generation of •$O_{2}^{-}$ and •OH radicals. Complementarily, the protonation of •$O_{2}^{-}$ species would generate H$O_{2}^{•}$ radicals, resulting in extreme instability over UV-illuminated H_2_O_2_, which could also give rise to hydroxyl •OH radicals [54]. The great involvement of active •OH radicals could lead to the easy formation of [OH-mono-chlorophenol]-intermediated species, which have been reported to be a major step in 4-CP photocatalytic degradation [55]. When Ag^(0)^ is incorporated into the TiO_2_ nanocrystals through the defect twin boundaries, the interaction between oxygen and the electron cloud of the adjacent Ag atoms can positively shift the chemical state of Ag atoms and activate the subsequent oxidation reaction. Therefore, the presence of multiply twinned Ag^(0)^ on the surface of nanostructured TiO_2_, even when introduced by completely different synthetic routes, can provide a better opportunity for the adsorption and degradation of organic pollutants and can boost PEC activity.

## 4. Conclusions

In summary, the CLT and laser ablation techniques were used to form Ag_TiO_2_ nanosheets and Ag_TiO_2_ nanostructured thin films (NSTFs), respectively. The efficiency of both methods was compared in terms of morphology and photocatalytic activity. A common advantage concerning the morphology is a single multiply twinned Ag^(0)^ characterized by {111} twin boundaries. The Ag_TiO_LYO/800 nanosheets exhibited enhanced photocatalytic activity under UV light in the decomposition of 4CP and TOC followed by Ag_TiO_LYO/650. Ag_TiO_LYO/800 possessing the best photoactivity could be explained by the evenly distributed multiply twinned Ag^(0)^NPs with pentagonal bipyramidal geometry and the reduced *E_bg_* of 3.07 eV, compared with standard TiO_2__P25 and Ag_TiO_2_ nanosheets annealed at lower temperatures, and pristine 2D TiO_2_ nanosheets obtained by CLT. The Ag^(0)^ stabilized defect-free anatase (101) TiO_2_ grains at temperatures of up to 800 °C, which in turn may serve as nucleation sites for the growth of Ag^(0)^NPs with pentagonal bipyramidal geometries. Ag-TiO_2_ NSTFs decorated with multiply twinned Ag^(0)^ achieved improved photoelectrochemical water splitting in the visible light region due to an additionally induced plasmonic effect. The CLT and laser ablation technique methods supported the formation of immobilized multiply twinned Ag^(0)^ on the TiO_2_ surface, which was able to avoid (e−/h^+^) recombination and contribute to the generation of •$O_{2}^{-}$ and •OH radicals favoring the photocatalytic activity of Ag_TiO_2_ materials.

## Data Availability

MDPI Research Data Policies at https://www.mdpi.com/ethics, accessed on 10 January 2022.

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
