# Peer review of "Effect of Multiply Twinned Ag^(0)^ Nanoparticles on Photocatalytic Properties of TiO_2_ Nanosheets and TiO_2_ Nanostructured Thin Films"

_nanomaterials, 2022, doi:10.3390/nano12050750_

Round 1
Reviewer 1 Report
- In Scheme 1, the amounts of AgNO3 and TiOSO4 should be written correctly. The milliliter abbreviation should also be the same in sections 2.2.1 and Scheme-1.
- The authors adopted nanotechnology to fabricate the samples but did not write even a single sentence regarding the importance of nanostructured photocatalysts in the introduction section.
- It is important to describe the role of Ag(0) in the TiO2 photocatalytic activity.
- The font style, especially the headings, should be identical throughout the manuscript.
- In section 3.1, PDF and JCPDS numbers must be supported with the proper citations. The use of italic words in this section should be avoided.
- Using XRD data, the crystallinity should be calculated. For help, 272 (2021) 116645 article could be consulted.
- The quality of figures, especially Fig. 5, should improve. The degree symbol should be carefully checked throughout the whole article.
- All the commonly used formulas must be written (like Scherrer and others). Proper numbers should be given to the used equations and formulas.
- The cyclic activity of the fabricated photocatalyst should be tested.
Author Response
Please, see the attachment.

Reviewer 2 Report
This manuscript reports the loading of Ag(0) nanoparticles on TiO2 nanosheets and thin films via two distinct techniques. The Ag loading is believed to contribute to the photoactivity towards organic pollutant degradations and photocatalytic water splitting. The characterizations in the Ag deposits are sufficient. The manuscript could be accepted after a major revision considering the follows.
- Except the loading of multiply twinned Ag(0), the two synthetic routes are quite distinct and the nanosheets powders and thin films share little in common. The photocatalytic performance for two series of samples is also distinct. It is thus suggested to focus on one synthetic route only in a separate manuscript. Studies on thin films could be submitted as a separate paper, if the authors want to.
- The TEM images do not show 2D nanosheets clearly. Please provide SEM images to show the overall morphology. With crystallites larger than 35 nm, it is not easy to imagine the overall 2D morphology. How about the lateral size and thickness of the nanosheets?
- UV Vis spectroscopy as illustrated in Figure 6 is not a necessary; but rather, the UV-Vis Diffuse reflectance spectra should be provided for a precise evaluation on the band gap.
- “as determined from the XRD is 238.7 (anatase) vs 251.1 (rutile)”: in my opinion, the Sherrer equation is not reliable to evaluate the crystallize size beyond 100 nm.
- In Figure 1, four samples not distinctive.
- Please rationalize why the surface Ag(0) concentration increased with increasing annealing temperature for up to 800 °C?
- No data can be found to support the statement that “The highest TOC removal was achieved in the presence of Ag_TiO_LYO/800 with complete mineralization (100% TOC removal) taking place after 30 h of photocatalytic treatment.”
- To support that “The key role of multiply twinned Ag(0) NPs on photocatalytic properties of TiO2 nanosheets and TiO2 NSTFs”, it is important to include the performance of samples without the Ag(0) loading.
- Simply CV curves are not enough to characterize the PEC water splitting performance.
- Figure 4c, the arrows should be vertical to the parallel lattice.
Author Response
Please, see the attachment.

Round 2
Reviewer 2 Report
The revised version is acceptable.